# Fully Vacuum-Sealed Diode-Structure Addressable ZnO Nanowire Cold Cathode Flat-Panel X-ray Source: Fabrication and Imaging Application

**DOI:** 10.3390/nano11113115

**Published:** 2021-11-18

**Authors:** Chengyun Wang, Guofu Zhang, Yuan Xu, Yicong Chen, Shaozhi Deng, Jun Chen

**Affiliations:** 1State Key Laboratory of Optoelectronic Materials and Technologies, Guangdong Province Key Laboratory of Display Material and Technology, School of Electronics and Information Technology, Sun Yat-Sen University, Guangzhou 510275, China; wangchy75@mail2.sysu.edu.cn (C.W.); zhanggfu@mail.sysu.edu.cn (G.Z.); chenyc25@mail.sysu.edu.cn (Y.C.); stsdsz@mail.sysu.edu.cn (S.D.); 2School of Biomedical Engineering, Southern Medical University, Guangzhou 510515, China; yuanxu@smu.edu.cn

**Keywords:** flat-panel X-ray source, ZnO nanowire, cold cathode, diode structure

## Abstract

A fully vacuum-sealed addressable flat-panel X-ray source based on ZnO nanowire field emitter arrays (FEAs) was fabricated. The device has a diode structure composed of cathode panel and anode panel. ZnO nanowire cold cathodes were prepared on strip electrodes on a cathode panel and Mo thin film strips were prepared on an anode panel acting as the target. Localized X-ray emission was realized by cross-addressing of cathode and anode electrodes. A radiation dose rate of 10.8 μGy/s was recorded at the anode voltage of 32 kV. The X-ray imaging of objects using different addressing scheme was obtained and the imaging results were analyzed. The results demonstrated the feasibility of achieving addressable flat-panel X-ray source using diode-structure for advanced X-ray imaging.

## 1. Introduction

X-ray imaging has important applications in the fields of medical diagnosis, security screening, industrial non-destructive inspection, etc. [1,2,3]. The X-ray source, as the core component of X-ray imaging system, plays a key role in the X-ray imaging technology. Thermionic cathode X-ray sources are widely used today. However, they are bulky and slow to respond. A cold cathode X-ray source could solve these problems, as it has the advantages of compact size, instant on-and-off, and low power consumption. 

Due to their excellent field emission characteristics [4,5,6], one-dimensional (1D) nanomaterials such as carbon nanotubes (CNTs) and nanowires were explored as the cold cathode for X-ray source [7,8,9]. H. Sugie et al. prepared the first CNT cold cathode X-ray tube and realized X-ray transmission imaging [10]. Subsequently, pulsed [11,12,13,14], micro-focus [15,16,17,18,19], and miniaturized X-ray sources [20,21,22,23] based on CNT cold cathode were developed. In the meantime, cold cathode X-ray sources using other 1D nanomaterials were also reported [24,25].

However, traditional X-ray sources usually have a single focus spot. In a computed tomography (CT) imaging system, it is necessary to move the X-ray source and detector to perform imaging at multiple angles. The mechanical movement limits the imaging speed and could induce blur, which affects the ultimate imaging resolution. Some researchers used multiple discrete cold cathode X-ray sources to solve the problem. J. Zhang et al. reported a multi-beam X-ray imaging system, which comprises an array of 5 cold cathode X-ray tubes and can be scanned by control circuits [26]. Therefore, transmission images from different projection angles can be obtained statically. Compared to the conventional tomographic scanner, this system is free from mechanical motion and has shorter scan time. Other teams later reported using a large number of cold cathode X-ray sources to achieve CT imaging [27,28]. However, due to the single focus nature of the discrete cold cathode X-ray tube, a large working distance is still required. Furthermore, a large number of tubes is required to achieve high-resolution imaging, making the imaging system complicated.

The flat-panel X-ray source proposed in recent years can achieve large area X-ray emission from arrays of micro-X-ray sources, which can greatly shorten the imaging distance and improve the imaging quality. In the configuration of flat-panel X-ray source, arrays of a large number of dense micro-X-ray sources could be formed. Each micro-X-ray source can have a focus spot in micron meter, which is helpful for improving the imaging quality. The addressable flat-panel X-ray source is expected to further develop new imaging methods by controlling the localized emission of X-rays, thereby achieving low dose, fast, and conformal CT imaging [29,30]. Otto Zhou et al. reported a small-size multiple-pixel X-ray source device using CNTs cold cathode, which is composed of a 5 × 10 pixel array [31]. E.J. Grant et al. proposed a flat-panel X-ray source consisting of two-dimensional X-ray micro-pixels using CNTs or nitrogen-doped ultra-nanocrystalline diamond film (N-UNCD) cold cathode as the electron source [32]. The technical feasibility of the flat-panel X-ray source was verified through simulation and a 2 × 4 pixel N-UNCD field emitter arrays (FEAs) was designed, while no experimental report on the device was presented [33]. G. Travish et al. proposed a flat-panel X-ray source using spontaneous polarization in pyroelectric crystals to generate high fields and enhance the field emission from micropatterned tips to create a large array of electron beamlets. Based on this, they produced a prototype device based on an 8 × 8 metal tip array [34]. In contrast, quasi 1D nanowire cold cathode materials such as ZnO nanowires can be easily synthesized on a large area by the thermal oxidation method, making them an ideal cold cathode candidate for realizing a large area flat-panel X-ray source. Chen et al. reported a large-area diode structure flat-panel X-ray source with ZnO nanowire cold cathode and high contrast X-ray images were obtained [35,36]. Flat-panel X-ray sources up to 4 inches in diagonal size have been demonstrated [37,38]. The above-mentioned work demonstrates the feasibility of using ZnO nanowire cold cathode for flat-panel X-ray sources. Recently, Cao et al. realized a large-area addressable flat-panel X-ray source by using gate-structure ZnO nanowire FEAs, where pulsed and addressable X-ray emission was successfully achieved by applying voltages to the extraction gate and the projection X-ray imaging of objects was realized [39].

Aside from the gate structure device, the diode structure device can also achieve an addressing function. The diode structure device is easy to fabricate due to its simple structure. On the other hand, the X-ray source operates under high anode voltage (up to hundreds of kV) and high voltage induced flashover is inevitable. By avoiding using delicate microstructure, the diode structure is more robust than a gate-structure device when working at high anode voltages. However, in a diode structure X-ray device, high anode voltage is needed for generating X-ray, while in the meantime, the current is modulated by the anode voltage. Therefore, the disadvantage of the diode structure addressing device is that it requires a high-voltage driving circuit. In contrast, in a gate structure device, the emission current is modulated by the gate located near the cathode with a typical distance of several microns, resulting in a low driving voltage. Recently, with the development of compound semiconductor, high-voltage electronics have achieved dramatic progress, e.g., high voltage (up to decades kV), high speed (~ns) driving circuits using SiC high voltage transistor have been reported [40]. Therefore, it is no longer an obstacle to realize high-voltage and high-speed driving circuits and this paves the path for diode-structure addressable flat-panel X-ray source.

In this work, we fabricated a fully vacuum-sealed addressable diode structure ZnO nanowire cold cathode flat-panel X-ray source. The row, column, and single-pixel addressing capability was verified. The X-ray energy spectrum and radiation dose rate were obtained, and the X-ray imaging using this device with different addressing schemes was also demonstrated.

## 2. Device Structure and Experimental Details

Figure 1a shows the schematic of the diode-structure flat-panel X-ray source, which is composed of an anode panel and a cathode panel. The anode panel consists of 5 strip-shaped molybdenum thin film anode targets prepared on a glass substrate. The cathode panel consists of 5 strip-shaped indium tin oxide (ITO) electrodes prepared on a glass substrate. The anode and cathode electrodes are arranged perpendicularly to each other and isolated electrically by spacer, which is a 6 mm thick rectangular glass frame. A 5 × 5 ZnO nanowire cold cathode pixel arrays were prepared on the cathode electrode at the crossing position of the cathode and anode electrodes. The size of each pixel is 4.325 mm × 4.325 mm, which is composed of 145 × 145 ZnO nanowire circular patterns with a diameter of 5 μm and a center distance of 30 μm. By selectively applying high voltage between the anode and cathode electrodes, addressable emission of X-ray could be achieved. The exhaust tube and evaporation type getters (barium aluminum nickel alloy) were installed on the cathode panel.

The fabrication process of the cathode panel is shown in Figure 2. Firstly, the substrate was cleaned with acetone, ethanol, and deionized water (see Figure 2a). Secondly, a 500-nm-thick ITO film was deposited on glass substrate by magnetron sputtering (see Figure 2b). Furthermore, cathode electrodes were etched in specific areas of the ITO film by wet etching combined with photolithography (see Figure 2c), and the etching solution used in the process is hydrochloric acid (HCl, concentration 36%) diluted with water (HCl:H_2_O = 1:5). After that, a 1400-nm-thick zinc thin film was deposited using electron beam evaporation deposition and patterned to circle-shape by a lift-off process (see Figure 2d). Finally, ZnO nanowires were obtained through a thermal oxidation process in an air atmosphere at 470 °C for 3 h (see Figure 2e). For the anode panel, a 400-nm-thick molybdenum (Mo) film was deposited on a 100-nm-thick aluminum (Al) film on a glass substrate, and another layer of an Al film with a thickness of 100 nm was deposited on top of the Mo film to prevent oxidization during the vacuum sealing process. Then, the Al-Mo-Al composite film was patterned to electrodes by using a photolithography and wet etching technique, where the etching was carried out in a mixed solution of phosphoric acid (H_3_PO_4_, concentration 85%), acetic acid (CH_3_COOH, concentration 99.5%), nitric acid (HNO_3_, concentration 65%), and water under 80 °C (H_3_PO_4_:H_2_O:CH_3_COOH: HNO_3_ = 24:12:6:1).

Figure 1b shows a picture of the fully vacuum-sealed addressable diode-structure flat-panel X-ray source that we fabricated. The process for vacuum encapsulation is described as follows. First, we used a low-melting point glass frit to seal the device. Secondly, the exhaust tube on the device was connected to the vacuum system which is equipped with a mechanical pump and a molecular pump. Thirdly, the device was pumped and at the same time, the device was baked under 300 °C. When the pressure reached 1 × 10^−6^ Pa, a ring-shape electrical heater, which is installed around the exhaust tube, was turned on to heat the exhaust tube. The exhaust tube was softened and sealed off. Finally, a radio frequency (RF) induction heater was used to activate the getters. The getters were heated and the barium in getters was evaporated and condensed on the inner surface of the exhaust tube, absorbing residual gas and maintaining the pressure inside the device.

The morphology of the cold cathode array of ZnO nanowires was characterized by scanning electron microscopy (SEM; Zeiss SUPRA 55, Jena, Germany). The field emission characteristics were measured using DC high voltage source and multimeter. The spectra of the generated X-rays were recorded by an X-123 CdTe X-ray spectrometer (AMPTEK, Bedford, MA, USA). The radiation dose was measured by an X-ray dosimeter (IBA MagicMax, Herndon, VA, USA). The X-ray images were captured by a flat-panel imaging detector (NDT0202M, iRay Technology, Shanghai, China).

## 3. Results and Discussions

The optical photograph of the cathode panel is shown in Figure 3a, in which the 5 × 5 arrays of rectangular pixel can be clearly seen. The morphology of the ZnO nanowires on the central pixel can be observed from Figure 3b. The statistical results of the morphology of the ZnO nanowires in the SEM image show that the average length, tip diameter, and growth density of the nanowires were 3.5 μm, 23 nm, and 4.3 μm^−2^, respectively. We also characterized the ZnO nanowires over the whole arrays. As can be seen from Figure 3c, the ZnO nanowires in various regions were uniformly distributed and had similar morphologies, which proves that ZnO nanowire field emitters were grown uniformly on the substrate.

The I–V characteristics of the device were measured when all the anode electrodes were applied with the voltage, and all the cathode electrodes were grounded. The result is presented in Figure 4. The result shows the emission current increased when the anode voltage increased. The current can reach a maximum of 2.7 μA when the anode voltage is 32 kV. The corresponding current density was calculated to be 26.2 μA/cm^2^. For a metallic emitter, the relationship between the field emission current I and the applied voltage V can usually be described by the Fowler–Nordheim (F-N) equation $I={\mathrm{AV}}^{2}\exp\left(-\frac{B}{V}\right)$, where A and B are constants [41]. It can be found that there is a linear relationship between 1/V and ln(I/V^2^). However, ZnO nanowire is an n-type semiconductor material. Strictly, the classical F-N formula does not apply. However, as it still lacks a simple equation to mathematically depict the field emission current from semiconductor nanowires, researchers still tend to use the F-N plot to analyze the field emission data. In fact, early field emission studies from semiconductor tip or nanowire arrays have shown that an approximately linear F-N plot with a negative slope could be obtained [42,43,44]. Therefore, the F-N plot is usually used to see if the current is from the electron emission. The inset of Figure 4 shows the corresponding F-N plot of the device, which shows a negative slope line demonstrating that the current is from electron emission.

The optical images were taken when the device was at work, which is presented in Figure 5. Apart from the X-ray generation, visible light emission could be observed when the electrons bombard the substrate. The addressing ability of the device was verified by observing the emission image. By applying high voltage to all anode cathodes and grounding all cathode electrodes, the full area emission image was obtained, which is shown in Figure 5a. By applying high voltage to all anode cathodes and grounding the first cathode electrode from the top, the row addressing emission image was obtained, which is shown in Figure 5b. By applying high voltage to the third anode electrode from left to right and grounding all cathode electrodes, the column addressing emission image was obtained, which is shown in Figure 5c. By applying high voltage to the third anode electrode from left to right and grounding the first cathode electrode from the top, the single-pixel addressing emission image was obtained, which is shown in Figure 5d. In the row addressing and single-pixel addressing experiments, the cathode electrodes for the pixels in the non-emission area were all connected with a voltage of 6 kV, in order to reduce the influence of crosstalk between electrodes. Our results indicate that the device exhibits good addressing ability.

The X-ray energy spectra were recorded by X-ray spectrometer when all pixels were applied with the anode voltages at 29, 30, and 31 kV. The typical X-ray energy spectra for anode voltage of 29, 30, and 31 kV are depicted in Figure 6. The spectra are mainly composed of bremsstrahlung X-ray emission, on which the characteristic X-ray peaks of Mo K-lines (K_α_ and K_β_) are superimposed. It is noticeable that the maximum energies of the X-ray signals in the spectra are consistent with the anode voltages. Furthermore, a maximum radiation dose rate of 10.8 μGy/s was obtained at an anode voltage of 32 kV as measured by the X-ray dosimeter.

X-ray transmission imaging of non-biological samples under different addressing schemes was obtained using the fabricated flat-panel X-ray source. Figure 7a,b shows the imaging results of the stainless-steel ring attached to the detector at a distance of 7.5 cm from the X-ray source, which were taken at a working voltage of 29 kV with an exposure of 10 s, where the X-ray source uses a single row emission addressing scheme. Figure 7c shows the imaging results of the circuit chip under the same condition, from which the outline of the chip pins can be well distinguished. It is also observed that the bottom and top edges of features are much sharper than the left and right edges in Figure 7a, and the pins in Figure 7c are much better resolved along the vertical other than the horizontal direction. This blur phenomenon is due to the extension of the illumination source in the horizontal direction.

Furthermore, we imaged the chip again when all rows of the X-ray source were turned on. The result is shown in Figure 7d. Compared with the previous imaging results under single row emission, the image becomes more blurred when all pixels emit X-rays, where the outline of the chip pins is almost indistinguishable. We believe that the blurring of the image is caused by the overlap of X-ray projections originating from the X-ray beams with different angles emitted from different rows [45].

We calculated corresponding edge spread function (ESF) and line spread function (LSF) using the image of the chip shown in Figure 7c. As shown in Figure 8a, we chose two locations corresponding to the chip pins in the horizontal and vertical directions, to calculate the LSF. We first obtained the corresponding profile in the horizontal and vertical directions (Figure 8b). Then, ESF and LSF were abstracted (Figure 8c,d). It can be seen that the profile in the vertical direction is more regular with lower noise. The ESF in the vertical direction falls faster than horizontal direction, which is consistent with the phenomenon observed in the picture. This shows that the vertical resolution is better than the horizontal direction. The LSF results are consistent with the ESF results.

We further measured this blurring effect under a different illumination scheme. We used the top single row (row B) and bottom single row (row A) of the X-ray source to emit X-ray separately to perform transmission imaging of a hexagonal nut. During the imaging process, the distance between the nut and the X-ray source was 7 cm, the anode voltage of the device was maintained at 29 kV, and the exposure was 20 s. The imaging results are shown in Figure 9a,b. It can be seen that the position of the nut image obtained from the row A (Figure 9a) was higher than that obtained from the row B (Figure 9b). We defined the relative displacement of the position as Δ_1_ and Δ_2_. The value of Δ_1_ and Δ_2_ was about 3.4 mm and 2.8 mm as measured from the image.

The result can be explained by the optics model of the X-ray imaging shown in Figure 9c. Since the electrodes have a certain width, the X-ray generated by a single row addressing emission will include X-ray beams in different directions. In order to simplify the analysis, we only plotted the X-ray beams from the edge of the pixel which define the outline of the object in the model. It can be clearly seen from the model that when row A was used for imaging, the corresponding imaging result was A’, and when row B was used for imaging, the corresponding imaging result was B’. Due to the shadow effect, a different displacement will occur when the pixels in row A or row B are turned on. The values of the displacement Δ_1_ and Δ_2_ is estimated from the X-ray optics, which is 3.2 mm and 2.3 mm, respectively. In the model, it is assumed that the surface of the flat-panel detector is 5 mm away from the actual semiconductor detector inside it. The calculation results show that both Δ_1_ and Δ_2_ are close to the experimental results and Δ_1_ > Δ_2_. The deviation in the value may be that the width of the X-ray beams emitted by a row is not completely consistent with the width of the electrode and the scattering of X-ray might also blur the edge of the image in the experiment. Overall, the experimental results confirmed that the device can achieve different-angle X-ray projection imaging through row addressing, which means that it is possible to use the device to project a multi-angle image of an object and then perform image processing to achieve conformal CT imaging. 

## 4. Conclusions

We developed a fully vacuum-sealed addressable diode-structure flat-panel X-ray source based on ZnO nanowire field emitter arrays. The effective production of X-ray was obtained from the device and the addressing capabilities have been confirmed. The maximum radiation dose rate of 10.8 μGy/s was obtained and corresponding X-ray transmission imaging of non-biological objects was achieved. Moreover, the multi-angle transmission imaging of the object was realized through the row addressing of the device. Our work demonstrates the feasibility of addressable diode structure cold cathode flat-panel X-ray source, which shows potential applications for novel X-ray imaging.

## Figures and Tables

**Figure 1 nanomaterials-11-03115-f001:**
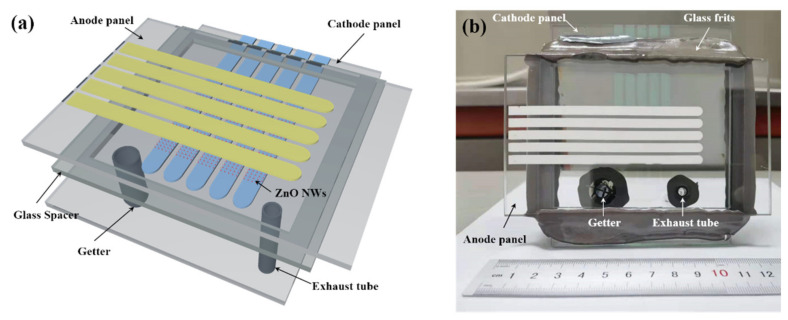
(**a**) Schematic diagram of the vacuum-sealed addressable diode-structure flat-panel X-ray source; (**b**) a picture of the fully vacuum-sealed addressable diode-structure flat-panel X-ray source.

**Figure 2 nanomaterials-11-03115-f002:**
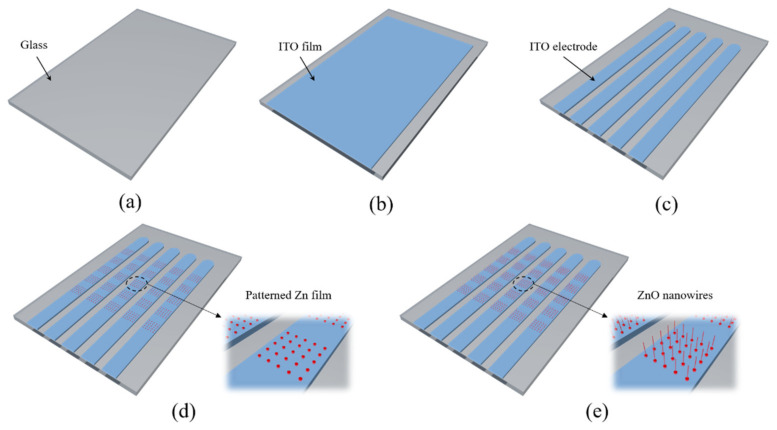
Preparation process of the cathode panel. (**a**) Cleaning of a glass substrate; (**b**) ITO film deposition; (**c**) etching of the ITO film to form electrode; (**d**) formation of Zn patterns; (**e**) growth of ZnO nanowires by the thermal oxidation method.

**Figure 3 nanomaterials-11-03115-f003:**
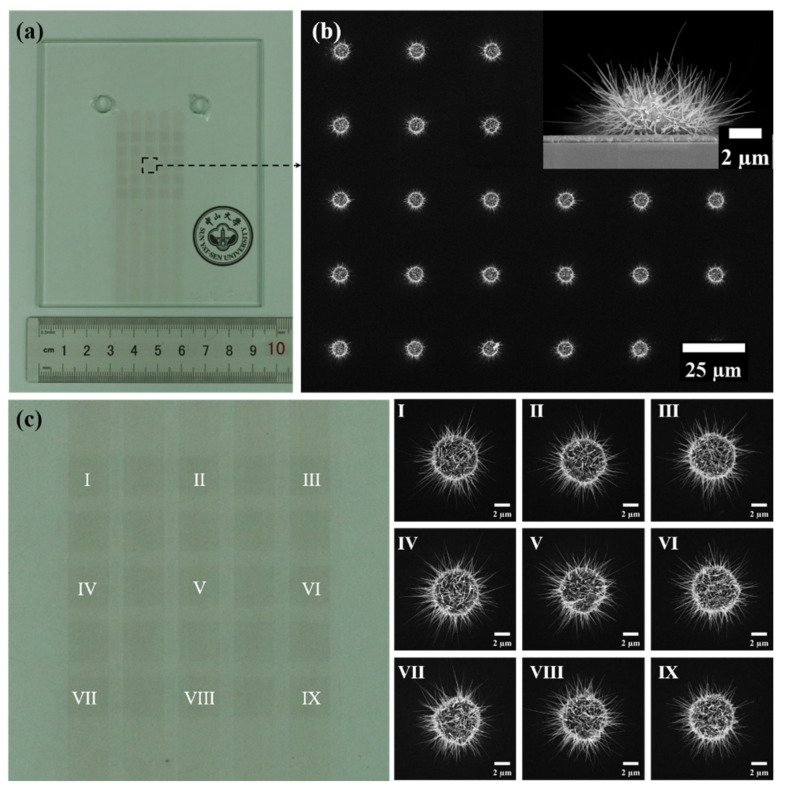
(**a**) Optical photograph of the cathode panel; (**b**) top view and cross-sectional view SEM images of the central pixel; (**c**) top view SEM images of 9 single patterns chosen from different areas.

**Figure 4 nanomaterials-11-03115-f004:**
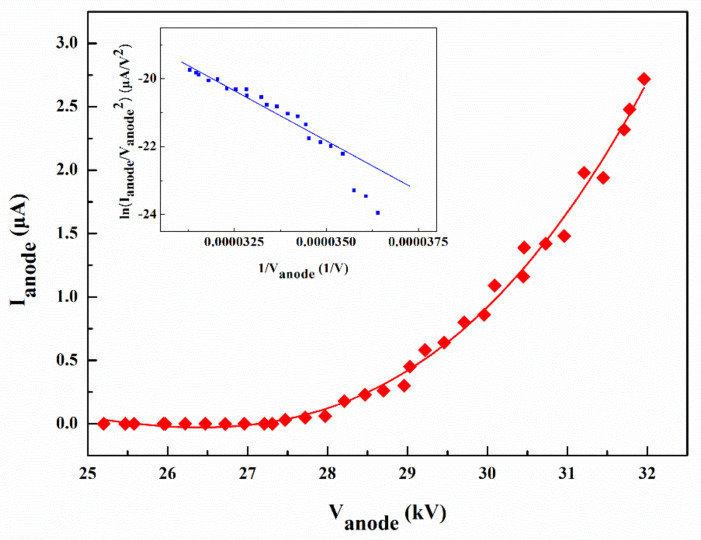
I–V curve of the flat-panel X-ray source and corresponding F-N plot (inset).

**Figure 5 nanomaterials-11-03115-f005:**
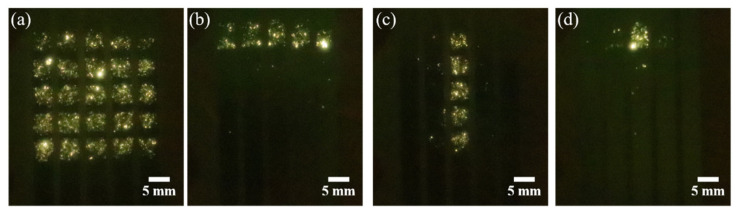
The photograph of visible light emission image at a voltage of 26 kV. (**a**) Full area emission; (**b**) row addressing emission; (**c**) column addressing emission; (**d**) single-pixel addressing emission. The images were taken by a Canon EOS 800D camera with ISO sensitivity, exposure time, aperture, and a focal length of 25,600, 1/4 s, f/4, and 100 mm, respectively.

**Figure 6 nanomaterials-11-03115-f006:**
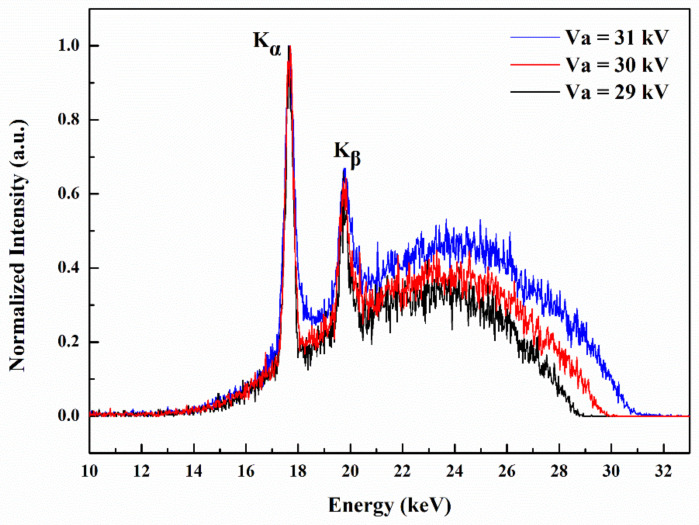
The normalized X-ray energy spectrum of the flat-panel X-ray source recorded under different anode voltages.

**Figure 7 nanomaterials-11-03115-f007:**
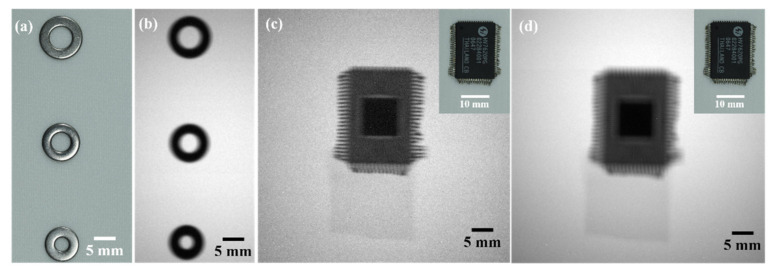
(**a**) Optical picture of stainless steel ring; (**b**) contact X-ray image of stainless steel ring obtained from single row emission of the device at 29 kV; (**c**) contact X-ray image of integrated circuit chip obtained from single row emission of the device at 29 kV, and inset shows the optical image; (**d**) contact X-ray image of integrated circuit chip obtained from all rows emission of the device at 29 kV, and inset shows the optical image.

**Figure 8 nanomaterials-11-03115-f008:**
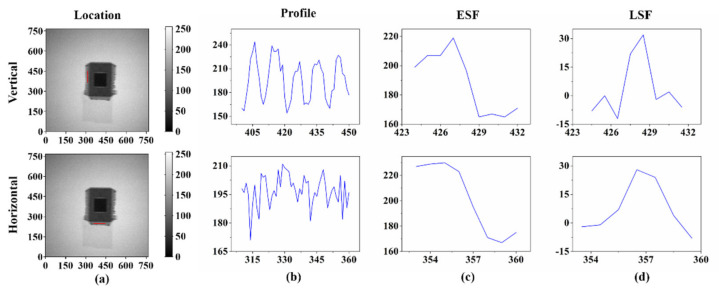
(**a**) Shows the two locations chosen from horizontal and vertical directions for data processing (see the red line). (**b**–**d**) Profile, edge spread function, and line spread function obtained.

**Figure 9 nanomaterials-11-03115-f009:**
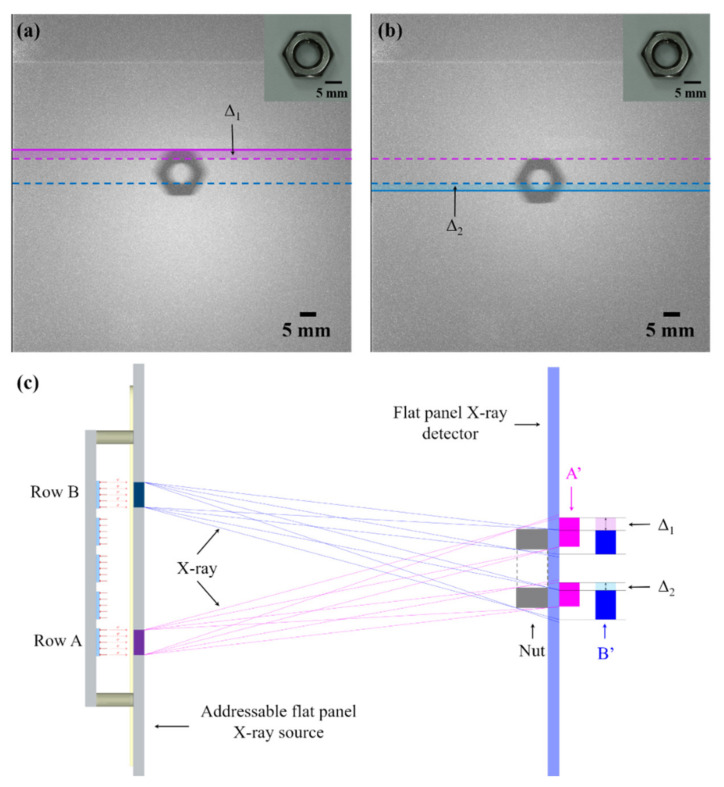
(**a**) X-ray image of nut obtained from the lowest row (row A) emission of the device at 29 kV, and inset shows the optical image; (**b**) X-ray image of nut obtained from the highest row (row B) emission of the device at 29 kV, and inset shows the optical image; (**c**) X-ray optics model from the emission of two row of the addressable flat-panel X-ray source.

## Data Availability

Data are contained within the article.

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
