# Peer review of "Fully Vacuum-Sealed Diode-Structure Addressable ZnO Nanowire Cold Cathode Flat-Panel X-ray Source: Fabrication and Imaging Application"

_nanomaterials, 2021, doi:10.3390/nano11113115_

Round 1

Reviewer 1 Report

The authors have built an interesting X-ray device. The impact of the results is sufficient for this journal, in my opinion. The text is well written and the results seem sound.

I only have a few comments:
* Why are the ZnO nanowires formed in 5x5 arrays, within each pixel?
* The 100 nm Al coating on the Mo seems quite thick - doesn't this reduce the electron beam significantly?
* Can the authors estimate the final UHV pressure?
* Please estimate the integrated X-ray flux of the source, and compare it with previous reports and commercial systems. 
* The parallax / source size blurring effect discussed on page 7 and 8 is well known. See e.g. : "T. E. Gureyev, Y. I. Nesterets, A. W. Stevenson, P. R. Miller, A. Pogany, and S. W. Wilkins, "Some simple rules for contrast, signal-to-noise and resolution in in-line x-ray phase-contrast imaging"  Opt. Express 16 (5), 3223 (2008) http://dx.doi.org/10.1364/OE.16.003223"

Reviewer 2 Report

The manuscript describes the fabrication and first application for X-ray imaging using an array of zinc oxide based nanowire field emitters. The manufacture and characterisation is interesting and suitable for the journal. 

Structure and figures are fine but the text will need some revision before it can be accepted. 

There are numerous grammar issues, mainly around incorrect use of plural/singular forms and a confusing mix of present tense and tense; these should be eliminated with the help of a native speaker or a professional translator. Sometimes incorrect words have been chosen, e.g. ’obtain statically’ in line 43 should be replaced by ‘obtained’, ‘idea’ in line 66 by ‘an ideal’, ‘from’ in line 149 by ‘in’, ‘a n-type’ in line 169 by ‘an n-type’, ‘since’ in line 170 by ‘as it’, ‘line’ in lines 188 and 198 by ‘column’ (which describes a vertical line of pixels).

Some technical descriptions would need more information: 
1. The statement in line 51 on micro-X-ray sources should be extended to explain why and how these are expected to improve imaging quality. Will also corresponding multiple detectors be necessary? 
2. The difference between ‘gate structure device’ (line 76) and ‘diode structure’ (line 80) should be explained, either by a brief paragraph or by a suitable sketch so that the reader can understand why only the latter seem to need a kV bias. 
3. The authors should describe what the ‘spacer’ mentioned in line 99 is made of.
4. Some process parameters are missing, eg. the HCl acid concentration (line 115), anneal time (lines 118/119) and acid mixture ratios (lines 124/125). 
5. The procedure of sealing the device would have been clearer without the phrase ‘due to the pressure’ in line 133. What is the ‘chemical substance’ in the’ ‘getters’ (line 135) and how can they be ‘activated’?
6. In the cross-sectional view of Fig 3b the bushel of ZnO nanowires seems to hover in mid air above the substrate - why does there seem to be no connection?
7. The X-ray spectrum in Fig.6 shows only K-lines from Mo; I would have expected also lines from Si, O (glass!), Zn, In and Sn (ITO!). Why are the latter absent?
8. The authors use the parallax effect based on imaging using top and bottom row only to explain contrast blur, but this can indeed be explained at least equally well from Fig7 alone: the left and right edges of features are much sharper than the the bottom and top edges in Fig 7a, and the device contacts in Fig.7c are much better resolved along horizontal than vertical direction, which must be due to the extension of the illumination source in the vertical direction, ie. using a column instead of a line (certainly NOT ‘a single row’ as stated in line 226). From these images, the authors could have extracted a point spread function and a line spread function for their imaging device.  
